# Clinical Utility of Oncuria™, a Multiplexed Liquid Biopsy for the Non-Invasive Detection of Bladder Cancer—A Pilot Study

**DOI:** 10.3390/diagnostics12010131

**Published:** 2022-01-06

**Authors:** Kaoru Murakami, Ian Pagano, Hideki Furuya, Timothy Daskivich, Dave Mori, Charles J. Rosser

**Affiliations:** 1Cedars-Sinai Medical Center, Samuel Oschin Comprehensive Cancer Institute, Los Angeles, CA 90048, USA; kaoru.murakami@cshs.org (K.M.); hideki.furuya@cshs.org (H.F.); 2Cancer Prevention and Control Program, University of Hawaii Cancer Center, Honolulu, HI 96813, USA; pagano@hawaii.edu; 3Cedars-Sinai Medical Center, Division of Urology, Los Angeles, CA 90048, USA; Timothy.Daskivich@cshs.org; 4Nonagen Bioscience Corporation, Los Angeles, CA 90010, USA; mori@nonagen.com

**Keywords:** biomarker, clinical utility, cytology, cystoscopy, hematuria, bladder cancer, molecular diagnostic, urine

## Abstract

Oncuria™ is a validated quantitative multiplex immunoassay capable of detecting bladder cancer from a voided urine sample. Herein, we sought to determine whether Oncuria™ affects physicians’ use of non-invasive and invasive diagnostic tests for microhematuria, gross hematuria, and bladder cancer surveillance. We conducted a survey-based study to assess physician management of nine clinical scenarios involving real-world data from patients with gross hematuria, microhematuria, and bladder cancer on surveillance. We randomly sampled 15 practicing urologists and generated data including 135 patient-by-urologist interactions and 2160 decision points. Urologists recommended a selection of diagnostic tests and procedures before and after Oncuria™ results were provided. We assessed changes in provider use of non-invasive and invasive diagnostic tests after Oncuria™ results were provided. Over 90% of all urologists changed their diagnostic behavior in at least one patient case with the addition of Oncuria™ results. The total number of diagnostic procedures was reduced by 31% following the disclosure of a negative Oncuria™ test and 27% following the disclosure of a positive Oncuria™ test. This is pilot study has the potential to shed light on the analysis of our four large multicenter international studies deploying Oncuria^TM^. The Oncuria™ urine-based test, a molecular diagnostic capable of ruling out the presence of bladder cancer, reduces both unnecessary invasive and non-invasive diagnostics and has the potential to reduce costs and improve patient outcomes.

## 1. Introduction

With the adoption of high-throughput technologies along with sequencing of the human genome almost 20 years ago [1], there has been an explosion of biomarker discovery and validation studies, leading to complex molecular diagnostic tests used to diagnose and stage cancers, to guide therapeutic selection, to assess treatment response, and to detect residual or recurrent cancer [2]. Despite all of this, the integration of molecular diagnostic tests into clinical practice has been inefficient largely due to test variability, a lack of analytical validation, limited robust clinical validation studies, as well as challenges with reimbursements. To address these limitations, the ACCE framework (Analytic Validity, Clinical Validity, Clinical Utility and Ethical, Legal and Social Implications) has been proposed to allow for the collection, evaluation, interpretation, and reporting of key molecular data associated with modern molecular diagnostic tests [3]. 

Utilizing the ACCE framework, we previously reported on the analytical validity of Oncuria™, a quantitative multiplex immunoassay capable of evaluating voided urine samples for the following biomarkers: angiogenin (ANG), apolipoprotein E (APOE), alpha-1 antitrypsin (A1AT), carbonic anhydrase 9 (CA9), interleukin-8 (IL-8), matrix metallopeptidase 9 (MMP9), matrix metallopeptidase 10 (MMP10), plasminogen activator inhibitor 1 (PAI1), syndecan 1 (SDC1), and vascular endothelial growth factor (VEGF), all of which comprised a validated bladder-cancer-associated diagnostic signature [4]. Subsequently, a clinical validation study reported an area under the receiver operating characteristic curve (AUROC) of 0.93 (95% CI: 0.87–0.98), outperforming any single biomarker in the signature [5]. The addition of critical demographic data (i.e., age, gender, and race) improved diagnostic performance to an AUROC of 0.95 (95% CI: 0.90–1.00) with an overall sensitivity of 0.93, specificity of 0.93, positive predictive value (PPV) of 0.65, and negative predictive value (NPV) of 0.99 for bladder cancer classification [6]. 

Clinical utility studies—which test whether the information from a diagnostic test leads to a change in patient management and improved health outcomes [7]—are an often overlooked, unmet need in molecular diagnostic testing. In fact, only two clinical utility studies have been performed on new molecular urine-based tests for the detection of bladder cancer [8,9]. Focusing on detection, prognostic, and predictive tests, there are several examples of these molecular diagnostic tests that are currently used in clinical practice to risk stratify cancer patients and target interventions, with accompanying evidence that the use of these tests leads to improved health outcomes for patients (e.g., Oncotype Dx™ and BRACAnalysis™) [10,11]. Nevertheless, there exists a large group of potentially promising molecular diagnostic tests that currently lack adequate evidence of clinical utility. Consequently, practice guideline committees and payers evaluating these tests often conclude that there is insufficient evidence to recommend clinical use or coverage, which correspondingly limits patient access. In the present study, we sought to demonstrate the clinical utility of Oncuria™ by investigating its impact on diagnostic decisions made by practicing urologists on real-world patients with hematuria (gross or microscopic) or bladder cancer surveillance being evaluated for bladder cancer. 

## 2. Materials and Methods

Practicing urologists in the US were asked to review 9 clinical vignettes: 3 related to the evaluation of gross hematuria, 3 related to the evaluation of microscopic hematuria, and 3 related to the monitoring of patients with a history of bladder cancer (Figure 1 Study Schema). Eighty-five practicing urologists selected at random from the American Urological Association physician registry were sent an email invitation describing the nature of the study and requesting participation in this prospective study. Within the email was a link to a survey monkey questionnaire (Appendix A), which collected general demographic data. By completing the survey monkey questionnaire, the physician provided consent of their participation in the study. The first 15 respondents were then selected to participate in individual virtual Zoom interviews with a trained moderator, who would present the 9 real-world clinical vignettes. Appendix A denotes the demographics of the participating urologists, which were generally reflective of national demographics [12]. The moderator took detailed notes of each encounter (quantitative and qualitative) with a focus on each participant’s responses to the questions. Prior to the Zoom interview, participants were asked to read two recently published articles describing the analytical validation of Oncuria™ [4] and the clinical validation of Oncuria™ [6] to provide context to some of the questions in the clinical vignettes (Appendix A). 

Each participant evaluated the same 9 clinical vignettes. These scenarios included actual clinical referral data from patients presenting with these symptoms systematically selected from the database of patients enrolled in previous prospective clinical studies of Oncuria™, which reported the Oncuria™ risk score or a bladder-cancer-associated diagnostic signature, urinary cytology, and/or another urine-based bladder cancer diagnostic and that would help illustrate the performance of Oncuria™ in the prescribed setting. Each participant was asked to review and recommend urine-based bladder cancer detection tests as well as diagnostic procedures such as cystoscopy and imaging for each case based on the patient’s evaluation from normal referral data. For the purposes of this study and to enable a reduction in any systemic bias, participants were instructed to consider all tests and procedures to be fully covered, with no additional out-of-pocket costs to the patient. Next, the physicians were provided with the Oncuria™ data. At the conclusion of each vignette, participants were informed of the final disposition of the patient. The study was approved by the Cedars-Sinai Medical Center IRB (00001548).

### 2.1. Study Endpoints

The co-primary endpoints of the study were (1) changes in the total number of urine-based bladder cancer detection tests used and (2) the number of invasive and/or radiologic procedures used before and after the disclosure of Oncuria’s™ diagnostic information. 

### 2.2. Heatmap

Data were graphically presented as heatmap calculations to aid in the visualization of the results at the individual participant–patient interaction level. The heatmaps show the range of variation in decision making and the change in decisions with and without the addition of the Oncuria™ result. The baseline heatmaps provide the total count of procedures at each of the 405 physician–patient interactions associated with the evaluation of patients with gross hematuria, microscopic hematuria, and bladder cancer surveillance (Figure 2). The change, relative to this baseline, made when the Oncuria™ result was disclosed reflects the change in participants’ decision making. Heatmap columns represent individual patient scenarios, and rows represent individual participants, with each row and column intersection becoming a participant–patient interaction. 

### 2.3. Statistical Procedures

Mixed (repeated measures) Poisson regression models were used to assess the change in the number of physicians recommending a procedure after the Oncuria™ result was disclosed. Separate models were constructed for each procedure, each procedure type (invasive, non-invasive, urinary, and non-urinary, all), and each Oncuria™ result (positive, negative, and all). The SAS 9.4 GENMOD procedure (SAS Institute Inc., Cary, NC, USA) was used to perform the analyses.

## 3. Results

Fifteen participating urologists evaluated nine individual patient vignettes (three with gross hematuria, three with microscopic hematuria, and three with a history of bladder cancer on tumor surveillance). Each patient encounter had eight questions related to the current use of diagnostics as well as the use of the same diagnostics with the Oncuria™ risk score available. Overall, there were 2160 participant–patient clinical decisions calculated as follows: 15 physicians (N_P_) × 9 vignettes (N_V_) × 8 questions (N_q_) × 2 assessments (with and without Oncuria results) (N_A_). The most recommended urine-based test to evaluate gross hematuria, microscopic hematuria, and bladder cancer surveillance was cytology in 31/45 (69%), 26/45 (58%), and 37/45 (82%), respectively (45 = 15 physicians × 3 scenarios: gross, microscopic and surveillance). Furthermore, the most recommended invasive procedures to evaluate gross hematuria, microscopic hematuria, and bladder cancer surveillance were office cystoscopy and CT scan in 43/45 (96%), office cystoscopy and CT scan in 34/45 (76%), and office cystoscopy in 26/45 (58%) (Figure 3). 

Over 90% of all urologists changed their diagnostic behavior in at least one patient case after being informed of the Oncuria™ results. Following a negative result for Oncuria™, there was a net reduction in procedures recommended in 34/56 (61%) of clinical decisions where additional diagnostic procedures were previously requested (Figure 2A and Table 1). Specifically, negative results for Oncuria™ resulted in the following reduction in the use of urine-based tests: 23/41 (56%) cytology, 7/9 (78%) Urovysion, and 4/4 (100%) Bladderchek. Similar reductions in invasive diagnostics were noted: 8/58 (14%) cystoscopy, 4/39 (10%) CT scan, and 3/6 (50%) ultrasound (Figure 2B and Table 1).

Following a positive result for Oncuria™, there was a net reduction in procedures recommended in 53/77 (69%) of clinical decisions where additional diagnostic procedures were previously requested (Figure 2A and Table 1). Specifically, positive results for Oncuria™ resulted in the following reduction in urine-based tests: 35/53 (66%) cytology, 13/16 (81%) Urovysion, and 5/5 (100%) Bladderchek. Similar reductions in invasive diagnostics were noted: 10/73 (14%) cystoscopy and 4/59 (7%) CT scan (Figure 2B and Table 1). Notably, with a strongly positive Oncuria™ test (i.e., risk score >70%), 9/90 (10%) of decision points noted participants to be amenable to proceeding directly to a transurethral resection of bladder tumors (green cells), which is the definitive diagnostic procedure. This is the only time a test was added after the results of Oncuria™ were revealed. However, this was only added as a consequence of eliminating office cystoscopy. 

## 4. Discussion

In this survey study, we found that Oncuria™, a validated quantitative multiplex immunoassay for the non-invasive detection of bladder cancer, reduced the total number of procedures requested by physicians, including reductions in both the number of urine-based tests and invasive procedures requested. We created a heatmap visualization to show changes in decision making at the level of the individual participant–patient interaction. Over 90% of all urologists who participated in this study made clinically relevant changes to their diagnostic workflow following the addition of Oncuria™ data. The study also identified a wide variation in the clinical diagnostic algorithm between physicians [13], which should be taken into consideration as diagnostics come onto the market. Interestingly, in patients with a strongly positive Oncuria™ test, 10% of urologists were more amendable to bypassing the office cystoscopy and going directly to the definitive diagnostic procedure of a transurethral resection of the bladder tumor to expedite patient care in patients with gross hematuria or bladder cancer surveillance and in so doing reducing costs and perhaps expediting evaluation and treatment. 

Urinary cytology, first reported in the literature in 1945 [14], is the only established urine-based diagnostic test used in routine clinical practice for the diagnosis and surveillance of bladder cancer [15]. Cytology has a high specificity for high-grade tumors but is limited by low sensitivity and variable performance across centers, in addition to a poor accuracy for low-grade tumors [16,17]. Thus, cytology is not an ideal biomarker, yet it continues to be utilized in the US, with roughly 1.5M tests performed annually [18]. With Medicare reimbursement rates for cytology being approximately USD 500 (CPT code 88173 = USD 245 and CPT code 88305 = USD 252) [19], the US healthcare system spends nearly USD 750M per year on cytology. With the advent of the gene expression array, followed by proteomics, RNA sequencing, DNA sequencing, and metabolomics, it is rather surprising that few multiplexed diagnostics tests have been applied to the care of patients with bladder cancer. Our study notes the willingness of urologists to adopt a replacement (i.e., urologists would abandon cytology in over 65% of scenarios) if the replacement was noted to possess favorable performance characteristics, i.e., a high sensitivity and NPV. Furthermore, urologists noted that the risk score associated with Oncuria™ allowed it to be a more “actionable” test. Here, actionable is consistent with many payers’ concepts of a “medically necessary” test [20], which can entail consideration not only of the impact of the test on patient management, but also comparative effectiveness with the current standard of care. 

Urine-based assays have an obvious advantage for the detection of bladder cancer, as the urine is in direct contact with the tumor. Most importantly, urine may be non-invasively collected. In addition, the ease of the collection of urine ensures patient compliance and allows for copious sample collection and repeat sampling. Recently, others have noted the importance of urine as a liquid biopsy for bladder cancer [21,22]. 

This is not the first time we have reached out to practicing urologists for their input on urine-based diagnostics. Previously, we conducted focus groups to ask urologists what performance characteristics they needed to have in a bladder cancer diagnostic. Obviously, sensitivity was required, but additionally, they overwhelmingly reported a high NPV, i.e., they did not want to miss detecting cancer. Thus, with that said, when we derived our algorithm, we kept this in mind. Previously, we reported a PPV of 0.65 with the test and an NPV of 0.99. Our present results show that the net reduction in diagnostics/procedures was noted 61% of the time after a negative Oncuria™ result, while a net reduction in diagnostics/procedures was noted 69% of the time after a positive Oncuria™ result. One would think that the net reduction in diagnostics/procedures should be higher after a negative test result than after a positive test result since a negative test result gives more certainty. In a larger study, this may be the case. However, our study has several limitations. First, our study sample of 15 randomly selected urologists may not be representative of a national sample of urologists. Second, the current clinical vignettes presented in this study have not been tested for validity and reliability. Third, there is a limitation when aggregating data on disparate clinical scenarios. Despite these limitations, the information gathered in this study will be quite useful to direct future studies. Such future studies could focus on the longitudinal assessment of changes in the Oncuria™ risk score, as such changes may give insight into the risk associated with the development of bladder cancer. Indeed, a movement toward risk-based counseling and decision making holds the promise of personalizing medicine in a heterogeneous and complex clinical entity such as bladder cancer. 

With its high NPV, Oncuria™ is effective in ruling out patients who have a low probability of harboring bladder cancer. The present study represents a population-level analysis on the impact of employing Oncuria™ on the treatment decisions amongst urologists evaluating patients with hematuria (gross and microscopic) and a history of bladder cancer on tumor surveillance. As seen in this study, participating urologists demonstrated a wide variety of diagnostic decisions, which may be influenced by the patient’s clinical presentation. In particular, in this study, Oncuria™ provided greater clinical resolution, allowing urologists to define a final clinical diagnostic algorithm to best manage each patient’s disease. Additional information derived from Oncuria™ reporting led many urologists to adjust their decision making, specifically opting for less diagnostics/procedures, which may lead to better outcomes by expediting and streamlining care. This will be evaluated in our ongoing prospective trials. 

Currently, three large international multicenter clinical studies are underway to validate Oncuria’s™ ability to detect bladder cancer in patients with hematuria (NCT03193541 and NCT03193528), and its ability to detect recurrent bladder cancer in patients with a history of bladder cancer on tumor surveillance (NCT03193515). A secondary analysis of these studies will consider the cost benefit (weighing costs against all consequences of intervention in monetary units) and cost effectiveness (considers both costs in monetary units and outcomes presented in non-monetary units, e.g., prevented intervention) of implementing the new test compared with the current standard of care, cytology. We hypothesize that the incremental cost-effectiveness ratios (ICERs) for cytology compared with Oncuria™ will be extremely high since Oncuria™ has both high specificity and sensitivity for the detection of bladder cancer compared with cytology, with only a marginal increase in cost. 

## 5. Conclusions

In conclusion, we found that the Oncuria™ urine-based test—a highly promising actionable molecular diagnostic capable of ruling out the presence of bladder cancer--can reduce the number of unnecessary invasive and non-invasive diagnostics recommended by physicians, which has the potential to lower healthcare costs and improve outcomes for patients.

## Figures and Tables

**Figure 1 diagnostics-12-00131-f001:**
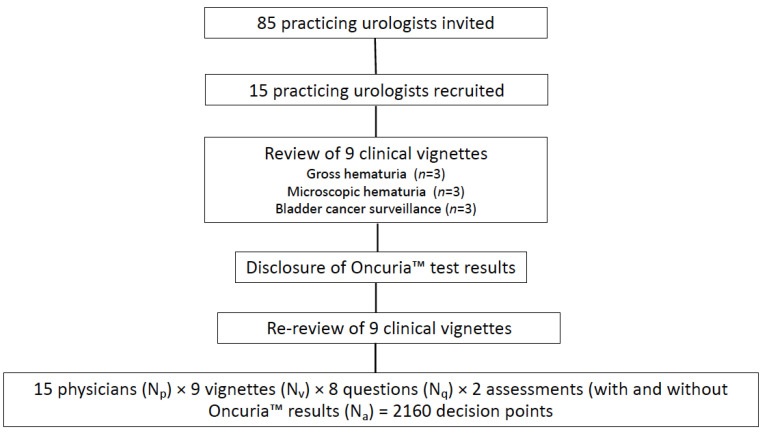
Study schema.

**Figure 2 diagnostics-12-00131-f002:**
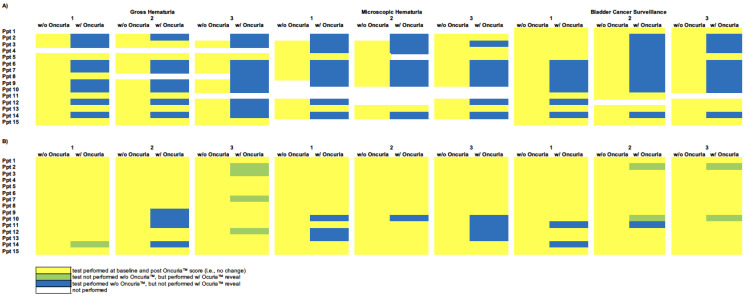
Heat maps representing the total number of diagnostic tests. This figure indicates the clinical decisions for the retrospective cases without (w/o) and with (w/) Oncuria™ results. Panel (**A**) baseline number of urine-based procedures and changes from baseline after presenting the results of Oncuria™. Rows represent participant physicians. Columns represent patients, w/o and w/Oncuria™ results. Each cell represents a patient–physician decision node. Yellow represents baseline decision and no change after disclosing Oncuria™ score. Green represents test not performed prior to Oncuria™ results disclosure but performed w/Oncuria™ results disclosure. Blue represents test performed prior to Oncuria™ results disclosure, but not performed w/Oncuria™ results disclosure. White indicates that no test was performed. Panel (**B**) baseline number of invasive and/or imaging procedures and changes from baseline after presenting the results of Oncuria™. Rows represent participant physicians. Columns represent patients, w/o and w/Oncuria™ results. Each cell represents a patient–physician decision node. Yellow represents baseline decision and no change after disclosure Oncuria™ score. Green represents test not performed prior to Oncuria™ data disclosure but performed w/Oncuria™ data disclosure. Blue represents test performed prior to Oncuria™ results disclosure, but not performed post-Oncuria™ results disclosure. White indicates that no test was performed.

**Figure 3 diagnostics-12-00131-f003:**
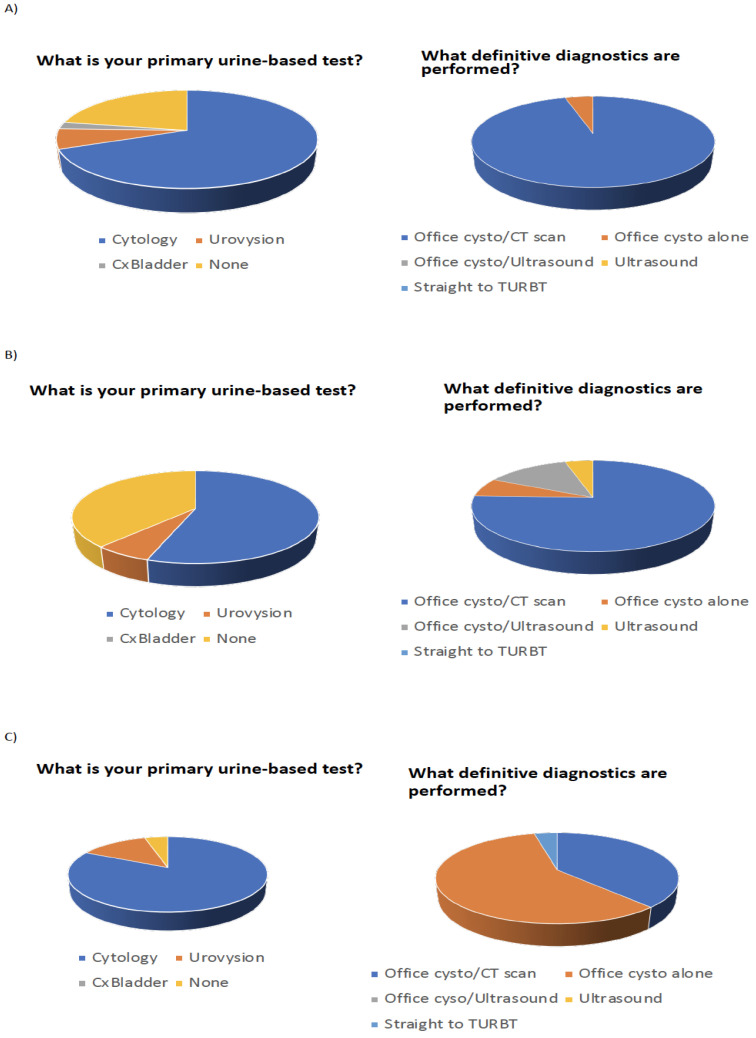
Pie chart representing the total number of primary diagnostic (invasive and non-invasive) tests performed for the evaluation of (**A**) gross hematuria, (**B**) microscopic hematuria, and (**C**) bladder cancer surveillance.

**Table 1 diagnostics-12-00131-t001:** Mean absolute and proportional change in the use of diagnostic tests (non-invasive and invasive).

			TOTAL (N_V_ = 9)						NEGATIVE (N_V_ = 4)							
	N_T_	N_P_	Pre	Post	Δ%	LCL	UCL	P	Pre	Post	Δ%	LCL	UCL	P	Pre	Post
Total	8	15	5.2	3.7	−29	−31	−26	<0.0001	5.0	3.4	−31	−31	−30	<0.0001	5.3	3.9
Invasive (I)	3	15	8.6	7.9	−7	−11	−4	<0.0001	8.1	7.1	−12	−16	−9	<0.0001	8.9	8.6
Non-Invasive (N)	5	15	3.1	1.1	−64	−69	−58	<0.0001	3.1	1.3	−60	−67	−51	<0.0001	3.2	1.0
Diagnostic (D)	4	15	6.6	6.1	−8	−12	−4	<0.0001	6.4	5.5	−15	−16	−13	<0.0001	6.8	6.6
Urine (U)	4	15	3.7	1.3	−65	−71	−59	0.0000	3.5	1.4	−61	−69	−50	0.0000	3.9	1.2
Bladderchek (NU)	1	15	1.0	0.0	−100				1.0	0.0	−100				1.0	0.0
CT Scan (ID)	1	15	10.9	10.0	−8	−12	−5	0.0000	9.8	8.8	−10	−15	−5	0.0001	11.8	11.0
Cxbladder (NU)	1	15	0.6	0.6	0				0.5	0.5	0				0.6	0.6
Cystoscopy (ID)	1	15	14.6	12.6	−14	−19	−8	<0.0001	14.5	12.5	−14	−21	−6	0.0015	14.6	12.6
Cytology (NU)	1	15	10.4	4.0	−62	−68	−54	0.0000	10.3	4.5	−56	−65	−45	0.0000	10.6	3.6
TURBT (ID)	1	15	0.2	1.2	450	109	1348	0.0006	0.0	0.0	0				0.4	2.2
Ultrasound (ND)	1	15	0.9	0.6	−38	−57	−10	0.01	1.5	0.8	−50	−50	−50	<0.0001	0.4	0.4
Urovysion (NU)	1	15	2.8	0.6	−80	−91	−56	0.0001	2.3	0.5	−78	−94	−21	0.02	3.2	0.6

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
