# Peer review of "Clinical Utility of Oncuria™, a Multiplexed Liquid Biopsy for the Non-Invasive Detection of Bladder Cancer—A Pilot Study"

_diagnostics, 2022, doi:10.3390/diagnostics12010131_

Round 1

Reviewer 1 Report

It's ok.

Author Response

Thank you very much for your comment. We are glad that you gave us a green light on previous version of manuscript.

Reviewer 2 Report

I thank the authors to consider my remarks.  My second remark was handled appropriately, but I still have a problem with Figure 2.  White means that no test was performed.  What kind of test are we speaking here about ?  Is it  a lab test ? In this case,  what is the difference between a test that is not ordered and a test that is not performed ? Is it maybe the Oncuria test itself ?  The authors should be more clear about this.

Author Response

We thank you for your review and kind comments. Below is a point-by-point response to your comments, so please kindly review the response as well as revised manuscript.

Comment 1

I thank the authors to consider my remarks.  My second remark was handled appropriately, but I still have a problem with Figure 2.  White means that no test was performed.  What kind of test are we speaking here about? Is it a lab test?

Response 1

Thank you for your understanding and raising this important comment.

We evaluated in figure 2. whether Oncuria test result can affect the further test, such as urine-based test and invasive test.

The details of urine-based test are shown in table 1. Urine-based procedures mean both urine-based and non-invasive test, including Bladderchek, Cxbladder, Cytology, and Urovysion. Invasive and/or imaging procedures include Ultrasound, CT scan, Cystoscopy,and TURBT. We added the explanation in figure legend so that readers can easily understand Figure 2. 

Panel A: Baseline number of urine-based procedures (Bladderchek, Cxbladder, Cytology, and Urovysion) and changes from baseline after presenting the results of Oncuria™.

Panel B: Baseline number of invasive and/or imaging procedures (Ultrasound, CT scan, Cystoscopy,and TURBT) and changes from baseline after presenting the results of Oncuria™.

Comment 2

In this case, what is the difference between a test that is not ordered and a test that is not performed?

Response 2

We are sorry for confusion about this point. A test that is not ordered and a test that is not performed is the same meaning. So we should have changed the yellow (test performed) to white (not performed), for example B), Gross Hematuria Patient 1, Ppt14, Pre-Oncuria. In addition, we changed ‘ordered’ to ‘performed’ to prevent confusion. According to your suggestion, we revised Figure 2.   

Comment 3

Is it maybe the Oncuria test itself? The authors should be more clear about this.

Response 3

 No, it isn’t. In this figure 2, we evaluated whether doctor’s decision making can change or not according to the Oncuria result. So Oncuria test was performed to all patients.

This manuscript is a resubmission of an earlier submission. The following is a list of the peer review reports and author responses from that submission.

Round 1

Reviewer 1 Report

The manuscript by Murakami et al described thea survey studyon the 
diagnostic performance of Oncuria™ urine-based test, of ruling out the 
presence of bladder cancer, reducing both unnecessary invasive and 
non-invasive diagnostics.

The study is interesting . However, the authors need to check 
throughout the text sevral grammar errors and need to edit language.
To further improve the manuscript, I suggest to add:
1)A schematic representation of the experimental approach used
2) a figure summarizing the conclusion of the study.
Moreover, discuss your findings on the basis of some recently 
published references regarding the relevance of liquid biopsy on the 
clinical management of patients with bladder cancer as the following:

-Ferro M, La Civita E, Liotti A, Cennamo M, Tortora F, Buonerba C, 
Crocetto F, Lucarelli G, Busetto GM, Del Giudice F, de Cobelli O, 
Carrieri G, Porreca A, Cimmino A, Terracciano D. Liquid Biopsy 
Biomarkers in Urine: A Route towards Molecular Diagnosis and 
Personalized Medicine of Bladder Cancer. J Pers Med. 2021 Mar 
23;11(3):237. doi: 10.3390/jpm11030237. PMID: 33806972; PMCID: 
PMC8004687.

-Huang HM, Li HX. Tumor heterogeneity and the potential role of liquid 
biopsy in bladder cancer. Cancer Commun (Lond). 2021 Feb;41(2):91-108. 
doi: 10.1002/cac2.12129. Epub 2020 Dec 30. PMID: 33377623; PMCID: 
PMC7896752.

-Crocetto F, Cimmino A, Ferro M, Terracciano D. Circulating tumor 
cells in bladder cancer: a new horizon of liquid biopsy for precision 
medicine. J Basic Clin Physiol Pharmacol. 2021 Sep 27. doi: 
10.1515/jbcpp-2021-0233. Epub ahead of print. PMID: 34563104.

Author Response

Reviewer #1

  1. The study is interesting. However, the authors need to check throughout the text several grammar errors and need to edit language.

Thank you for this comment.  We have had a medical editor extensively review and revise the manuscript.

  1. To further improve the manuscript, I suggest to add: 1) A schematic representation of the experimental approach used 2) a figure summarizing the conclusion of the study.

We have added a schematic (Fig 1) depicting our experimental approach. 

  1. Moreover, discuss your findings on the basis of some recently published references regarding the relevance of liquid biopsy on the clinical management of patients with bladder cancer as the following:

-Ferro M, La Civita E, Liotti A, Cennamo M, Tortora F, Buonerba C, 
Crocetto F, Lucarelli G, Busetto GM, Del Giudice F, de Cobelli O, 
Carrieri G, Porreca A, Cimmino A, Terracciano D. Liquid Biopsy 
Biomarkers in Urine: A Route towards Molecular Diagnosis and 
Personalized Medicine of Bladder Cancer. J Pers Med. 2021 Mar 
23;11(3):237. doi: 10.3390/jpm11030237. PMID: 33806972; PMCID: 
PMC8004687.

-Huang HM, Li HX. Tumor heterogeneity and the potential role of liquid 
biopsy in bladder cancer. Cancer Commun (Lond). 2021 Feb;41(2):91-108. 
doi: 10.1002/cac2.12129. Epub 2020 Dec 30. PMID: 33377623; PMCID: 
PMC7896752.

We have appropriately referenced these articles in the current version. 

Reviewer 2 Report

This manuscript reports the results of a study of the Oncuria system and assesses the influence of the Oncuria test result on the number and kind of diagnostic tests asked by the clinicianc.

I have two remarks: 
. * Figure 1:  I do not see the added value of showing the yellow columns for the Pre-Oncuria questions.  Only in those cases where a test was not ordered Pre-Oncuria but was Post-Oncuria, there is some added value, which could be solved by playing with, for example, four colors in the Post-Oncuria:  
color 1:  change from baseline
color 2:  tests not ordered pre-oncuria, tests ordered post-oncuria
color 3:  tests order pre-oncura, not ordered post-oncuria
color 4: no change from baseline

2. I read in the introduction that The PPV from Oncuria for bladder cancer detection is 0.65.  It means that in case of a positive test result, the patient has 65 % chance of actually having bladder cancer.
The NPV, on the contrary, is 0.99.  It means that in case of a negative test, there is almost absolute certainty that the patient is not developping bladder cancer. 
Conclusion:  a positive result should be handled with care (because there is 100-65) 45% chance of a positive test result, while negative test can be considered as very precise, since the chance of a false negative result is 1%.

The results show that the net reduction in procedures recommended is 61% after a negative Oncuria result, while it is 69% in case of a positive Oncuria result.  I am surprised about this, I would expoect that the net redouction on procedures should be higher after a negative test result than after a positive test result, since a negative test result gives more certainty.  

What is the opinion from the authors about this ?  I recommend adding this idea to the discussion.

Author Response

Reviewer #2

  1. Figure 1 I do not see the added value of showing the yellow columns for the Pre-Oncuria questions.  Only in those cases where a test was not ordered Pre-Oncuria but was Post-Oncuria, there is some added value, which could be solved by playing with, for example, four colors in the Post-Oncuria:
    color 1:  change from baseline
    color 2:  tests not ordered pre-oncuria, tests ordered post-oncuria
    color 3:  tests order pre-oncura, not ordered post-oncuria
    color 4: no change from baseline

This was revised as suggested.

  1. The results show that the net reduction in procedures recommended is 61% after a negative Oncuria result, while it is 69% in case of a positive Oncuria result.  I am surprised about this, I would expect that the net reduction on procedures should be higher after a negative test result than after a positive test result, since a negative test result gives more certainty.  What is the opinion from the authors about this?  I recommend adding this idea to the discussion.

This is a great question. As this is a survey administered to a small cohort of practicing urologists, we believe such subtly like a greater reduction diagnostic after a negative Oncuria™ may not bear out here, thus we anxiously await the results of our ongoing prospective studies. 

Reviewer 3 Report

In their manuscript entitled “Clinical Utility of a Oncuria™ a multiplexed liquid biopsy for the non-invasive detection of bladder cancer” Murakami and colleagues analyzed the performance of the Oncuria assay in a pseudo real-world setting. The study is well written and structured.

I have some points that should be addressed:

How were the examples selected from the database of patients enrolled in previous prospective clinical studies of Oncuria? A selection bias might blur the results of the study.

This is a survey study and no validation of the urologists’ choices based on the Oncuria assay results was given. Therefore, how can the authors be sure that the Oncuria-based decisions were correct compared to standard of care decisions?

In the discussion section, the results of the study are only very briefly discussed while a main emphasis is on reimbursement and costs issues. I think the manuscript would benefit from a more detailed discussion of the results of the study itself.

Author Response

Reviewer #3

  1. How were the examples selected from the database of patients enrolled in previous prospective clinical studies of Oncuria? A selection bias might blur the results of the study.

We systematically selected the cases for these vignettes from the database of patients enrolled in previous prospective clinical studies of Oncuria™, which reported the Oncuria™ risk score or the risk of harboring a bladder cancer, which reported urinary cytology and/or another urine-based bladder cancer diagnostic and that would help illustrate the performance of Oncuria™ in the prescribed setting. But you are correct. There is a potential for a selection bias.

  1. This is a survey study and no validation of the urologists’ choices based on the Oncuria assay results was given. Therefore, how can the authors be sure that the Oncuria-based decisions were correct compared to standard of care decisions?

In this setting, we had the results of standard of care diagnostics/test as well as the results of Oncuria™, which could be compared. As you mentioned earlier, they may be bias in the selection of these cases.  Also, the treating physicians in the vignettes did not have access to Oncuria to modify their decisions. However, for the ongoing prospective study, these issues will be addressed.

  1. In the discussion section, the results of the study are only very briefly discussed while a main emphasis is on reimbursement and costs issues. I think the manuscript would benefit from a more detailed discussion of the results of the study itself.

Thank you very much for this comment. We expanded the discussion to further discuss our results in the clinical context of its utility and adoption. 

Round 2

Reviewer 1 Report

It's ok after revisions.